# IDO1-Targeted Therapy Does Not Control Disease Development in the Eµ-TCL1 Mouse Model of Chronic Lymphocytic Leukemia

**DOI:** 10.3390/cancers13081899

**Published:** 2021-04-15

**Authors:** Selcen Öztürk, Verena Kalter, Philipp M. Roessner, Murat Sunbul, Martina Seiffert

**Affiliations:** 1Molecular Genetics, German Cancer Research Center (DKFZ), 69120 Heidelberg, Germany; s.oeztuerk@dkfz.de (S.Ö.); v.kalter@dkfz.de (V.K.); p.roessner@dkfz.de (P.M.R.); 2Institute of Pharmacy and Molecular Biotechnology, Heidelberg University, 69120 Heidelberg, Germany; msunbul@uni-heidelberg.de

**Keywords:** chronic lymphocytic leukemia, IDO1, epacadostat, immunotherapy, Eµ-TCL1

## Abstract

**Simple Summary:**

The tryptophan-catabolizing enzyme IDO1 and its metabolite kynurenine were shown to be enhanced in patients with chronic lymphocytic leukemia (CLL), and their involvement in T cell suppression and immune escape was suggested. As we have observed increased IDO1 expression and kynurenine serum levels in the Eµ-TCL1 mouse model of CLL, we evaluated the therapeutic potential of targeting IDO1 in preclinical treatment studies with two IDO1 inhibitors in mice developing CLL. As both studies revealed only minor effects of IDO1 inhibition on leukemia development and the immune compartment at early time points of treatment which disappeared over time, our data suggest that even though IDO1 might be involved in immunosuppressive mechanisms in CLL, its targeting is not sufficient for preventing immune escape. Thus, compensatory mechanisms beyond IDO1 seem to be of relevance to prevent clinically relevant benefits with IDO1-targeting drugs.

**Abstract:**

Indoleamine-2,3-dioxygenase 1 (IDO1), a tryptophan (Trp)-catabolizing enzyme producing metabolites such as kynurenine (Kyn), is expressed by myeloid-derived suppressor cells (MDSCs) and associated with cancer immune escape. IDO1-expressing monocytic MDSCs were shown to accumulate in patients with chronic lymphocytic leukemia (CLL) and to suppress T cell activity and induce suppressive regulatory T cells (Tregs) in vitro. In the Eµ-TCL1 mouse model of CLL, we observed a strong upregulation of IDO1 in monocytic and granulocytic MDSCs, and a significantly increased Kyn to Trp serum ratio. To explore the potential of IDO1 as a therapeutic target for CLL, we treated mice after adoptive transfer of Eµ-TCL1 leukemia cells with the IDO1 modulator 1-methyl-D-tryptophan (1-MT) which resulted in a minor reduction in leukemia development which disappeared over time. 1-MT treatment further led to a partial rescue of the immune cell changes that are induced with CLL development. Similarly, treatment of leukemic mice with the clinically investigated IDO1 inhibitor epacadostat reduced the frequency of Tregs and initially delayed CLL development slightly, an effect that was, however, lost at later time points. In sum, despite the observed upregulation of IDO1 in CLL, its inhibition is not sufficient to control leukemia development in the Eµ-TCL1 adoptive transfer model.

## 1. Introduction

Indoleamine-2,3-dioxygenases 1 and 2 (IDO1/2) and tryptophan-2,3-dioxygenase (TDO) are enzymes that catalyze the initial, rate-limiting step of converting the essential amino acid tryptophan (Trp) into kynurenine (Kyn). This metabolic activity was proposed to suppress T cell activity by its ability to deplete the inflammatory microenvironment of Trp [1,2]. IDO1 is the most studied enzyme of this family and according to current knowledge, it limits innate and adaptive immune responses by depleting immune effector cells of Trp [3,4], and by promoting the accumulation of Kyn and some of its derivatives which have been shown to induce T cell apoptosis and are involved in maintenance of peripheral T cell tolerance [5].

Upregulation of IDO1 either by tumor-infiltrating myeloid cells or by tumor cells themselves have been described in various malignancies and have been associated with a poor prognosis [6]. This led to the development of several IDO inhibitors which were shown to successfully suppress tumor formation in several animal models and which were tested in clinical trials for cancer patients [7]. However, most of the studies have been focused on solid tumors and an extensive analysis in lymphoid malignancies, especially in chronic lymphocytic leukemia (CLL), has not been performed to date.

An upregulation and activation of IDO1 in the CLL microenvironment has been reported. When serum from 49 patients with CLL and 24 age- and sex-matched controls was analyzed, the Trp to Kyn ratio was observed to be significantly, although moderately, increased [8]. In 2014, two independent studies have shown that monocytes from CLL patients show high IDO1 expression [9,10]. Furthermore, similar to findings in solid tumors, IDO-expressing myeloid cells from CLL patients have been shown to inhibit T cell responses and promote generation of regulatory T cells (Tregs) in vitro, suggesting that these cells represent myeloid-derived suppressor cells (MDSCs) [10]. In other lymphomas such as diffuse large B cell lymphoma (DLBCL), IDO1 expression and Kyn levels have been correlated with a poor prognosis [11].

CLL is associated with dysfunctional T cells that support the growth of leukemia cells and allow immune escape of this cancer [12]. Recent data suggest an initial CD8^+^ T cell-mediated immune control of CLL, which is lost due to their functional exhaustion [13]. The mechanisms that lead to this loss of tumor control are poorly understood. As there are indications that IDO1 contributes to the immunosuppressive microenvironment and T cell suppression in CLL, we investigated its role in the well-established Eµ-TCL1 mouse model of CLL.

## 2. Materials and Methods

### 2.1. Mouse Models

Adoptive transfer was performed by i.p. transplantation of 1–2 × 10^7^ leukemic Eµ-TCL1 splenocytes into respective, syngeneic animals to avoid tumor rejection [14]. Splenocytes were enriched for CD19^+^ cells using the EasySep™ Mouse Pan-B Cell Isolation Kit (Stemcell Technologies, Cologne, Germany), yielding a purity of above 95% CD5^+^ CD19^+^ cells.

All animal experiments were carried out according to institutional and governmental guidelines approved by the local authorities (Regierungspräsidium Karlsruhe, permit numbers: G123/14, G36/14, G98/16, G53/15). Eµ-TCL1 (C. Croce, OH, USA) mice were held in specific pathogen-free conditions on a pure C57BL/6 N or J background at the central animal facility of the German Cancer Research Center (DKFZ).

### 2.2. Collection of Tissue Samples and Preparation of Cell Suspensions

Mice were euthanized by increasing concentrations of carbon dioxide (CO_2_). Peripheral blood(PB) was drawn from the submandibular vein or via cardiac puncture and collected in ethylenediaminetetraacetic acid (EDTA)-coated tubes (Sarstedt, Nümbrecht, Germany). Single cell suspensions from spleens, bone marrow and inguinal lymph nodes (LNs) were prepared as previously described [15].

### 2.3. Flow Cytometry

Gating strategies are described in Appendix A. Bimodal populations were quantified as a percentage of protein-expressing cells and unimodal populations were analyzed as median fluorescence intensities (MFIs). Normalized MFI (nMFI) was calculated by subtracting the MFI of the respective fluorescence-minus-one (FMO) control.

Whole blood stainings were performed by the addition of the surface antibody cocktail to a defined volume of blood. Red blood cell lysis was performed by the addition of 1× Fix and Lyse Buffer (ThermoFisher Scientific, Dreieich, Germany) and subsequent incubation at room temperature. After centrifugation, the cell suspension was resuspended in PBS.

Single cell suspensions were stained in phosphate-buffered saline (PBS) with the addition of Fixable Viability Dye eFluor^®^ (ThermoFisher Scientific) at a concentration of 1:1000 for 30 min at 4 °C. Cells were fixed using IC Fixation Buffer (ThermoFisher Scientific) for 30 min at room temperature and kept at 4 °C in in dark until acquisition. For intracellular stainings, the eBioscience™ Foxp3/Transcription Factor Staining Buffer Set was used and the samples were stored at 4 °C in the dark until acquisition.

A list of antibodies is provided in Appendix A.

### 2.4. Determination of L-Tryptophan and L-Kynurenine Amounts in Mouse Serum

The reference metabolites (L-tryptophan and L-kynurenine) were purchased from Sigma-Aldrich (Taufkirchen, Germany) and used without further purification. Millimolar stock solutions of each standard were prepared in water and stored at −20 °C until use. Working solutions were freshly diluted from the stock solutions just before each measurement to create the HPLC standard curves. Reverse phase HPLC analysis was performed on an analytical column (Luna^®^ 5 µm C18(2) 100 Å, 250 × 10 mm, Phenomenex Ltd., Aschaffenburg, Germany) by using a mobile phase consisting of 50 mM acetic acid, 250 mM zinc acetate (pH 4.9) with 1% (v/v) acetonitrile. Separation of the peaks was achieved at 25 °C by isocratic elution at a flow rate of 1.5 mL/min. The detection and the quantification of L-tryptophan and L-kynurenine were carried out by monitoring their UV absorbance at 230 nm and 365 nm, respectively.

Fifty microliters of serum samples were mixed with 5 µL of 2.4 M perchloric acid on ice and briefly vortexed. After centrifugation (6000× *g*) at 4 °C for 15 min, the supernatants were transferred to a new tube, mixed with an equal volume of the mobile phase and filtered (0.22 µm Millipore filter). Forty microliters of the clear mixture were injected into the column. The concentrations of the metabolites in serum samples were determined by measuring the areas under the peaks and comparing them to the standard curve.

### 2.5. Treatment with 1MT and Epacadostat

For the treatment studies, a group size of 10 was chosen based on a simulation and on previous experience with this mouse model. For the epacadostat study, 20 mice were transplanted with tumor cells, but 4 mice had to be excluded from the experiment due to the failure of tumor cell engraftment. Therefore, the treatment was performed with 8 mice per group. As 2 mice from the vehicle and 1 mouse from the epacadostat group died during the course of treatment, they could not be included for the endpoint analysis.

1-methyl-D-tryptophan (1MT; Sigma Aldrich) was dissolved in 0.1 M NaOH and 0.25 mg/mL aspartame at a concentration of 2 mg/mL and added to the drinking water of mice with continuous access from the first day of tumor transplantation until the endpoint. Control mice received the vehicle (0.1 M NaOH and 0.25 mg/mL aspartame in drinking water). The dose and the route of administration were chosen according to previous studies showing the efficacy of 1MT [16,17,18].

Animals were treated by oral gavage every day with 4 mg epacadostat (INCB024360, MedKoo Biosciences, NC, USA) per mouse dissolved in 0.5% methylcellulose. Control mice received the vehicle (0.5% methylcellulose, Sigma-Aldrich, Taufkirchen, Germany). The dose and route of administration were recommended by Incyte Corporation and were similar to those described by Koblish et al., 2010 [19].

Treatment studies were terminated when the spleen size, determined by palpation, reached approximately 1.5 cm on average in any of the groups.

### 2.6. Statistical Analysis

Flow cytometry data were acquired using a BD FACS Canto II, BD LSR II or BD LSR Fortessa (BD Biosciences, Heidelberg, Germany) FACS analyzer and analyzed by FlowJo X 10.0.7 software (FlowJo, Ashland, OR, USA).

Data were analyzed using Prism 7 GraphPad software (GraphPad Software, La Jolla, CA, USA). Samples of different groups were compared using an unpaired *t*-test with Welch approximation to account for unequal variances. Values of *p* < 0.05 were considered statistically significant. All graphs show means ± SEM.

## 3. Results

Enhanced expression of IDO1 in monocytes and nurse-like cells from CLL patients was reported in two studies and was suggested to contribute to T cell suppression in this disease [10,20]. We analyzed spleen samples from primary Eµ-TCL1 (TCL1) mice with CLL-like disease by flow cytometry and observed elevated expression of IDO1 in MDSC-like classical monocytes and neutrophils in comparison to respective cells in wild-type (WT) mice (Figure 1A,B; Appendix A for gating strategy). This is in line with a previous study identifying monocytes with high IDO1 and low MHC II expression in the blood of CLL patients, cells that were described to act as MDSCs [10]. We further explored induction of IDO1 expression after adoptive transfer of TCL1 leukemia (TCL1 AT) in syngeneic WT mice and observed a higher frequency of IDO1-expressing monocytes and neutrophils in these mice compared to WT controls (Figure 1C,D). Interestingly, IDO1 expression was also significantly higher in malignant CD5^+^ CD19^+^ B cells of TCL1 mice compared to CD19^+^ B cells of WT mice (Figure 1E; Appendix A for gating strategy). In TCL1 AT mice, malignant CD5^+^ CD19^+^ B cells expressed significantly higher levels of IDO1 compared to the rare CD5^+^ CD19^+^ B cells from WT mice, which, interestingly, showed significantly higher IDO1 expression compared to conventional CD5^−^ CD19^+^ B cells in the same animals (Appendix A). To test the functional relevance of the increased IDO1 levels in the CLL mouse model, we analyzed Kyn and Trp levels in serum from leukemic end-stage TCL1 mice and their sex- and age-matched littermate mice by HPLC. The ratio of Kyn/Trp was significantly higher in TCL1 mice (Figure 1F), which is in line with previously described data showing moderately increased Kyn/Trp ratios in CLL patient serum [8]. Our data show that leukemia development in the TCL1 mouse model of CLL is associated with higher expression and activity of the Trp-degrading enzyme IDO1.

To investigate whether the high expression of IDO1 and concurrently elevated Kyn serum levels contribute to changes in the microenvironment and thereby impact on CLL development, we performed treatment studies in TCL1 AT mice using two compounds that were described to inhibit IDO1. Firstly, we analyzed effects of the IDO1 modulator 1-methyl-D-tryptophan (1MT) on tumor cell engraftment and leukemia development, whereby we started to treat the mice with 1MT on the same day they were transplanted with the tumor cells. 1MT treatment slightly reduced the expansion of malignant B cells in blood at an early time point, as observed by a lower percentage of leukemia cells in blood after 2 weeks of treatment with 1MT (Figure 2A). However, this effect disappeared in the following weeks and no differences in tumor load were detectable in blood after 3, 4 and 5 weeks of treatment. This lack of treatment efficacy was also reflected at the endpoint after 6 weeks of treatment when the tumor burden of spleen and lymph nodes was measured. Although there was a trend towards a reduced spleen size in 1MT-treated versus control mice, this difference was not significant and mainly caused by two animals with reduced spleen weights (Figure 2B). Similarly, the percentages of tumor cells in the spleen and inguinal lymph nodes were only slightly decreased, but no significant differences were observed (Figure 2C and Appendix A). When analyzing the proliferation rate of leukemic cells by KI67 staining, a strong trend towards a higher percentage of proliferating cells was detected in the spleen of 1MT-treated mice compared to controls (Figure 2D; Appendix A for gating strategy), which might explain the loss of treatment efficacy over time.

We further analyzed the impact of 1MT treatment on the tumor-associated immune cells in the spleen of these mice. Quantification of the main myeloid cell subsets revealed no major differences in the frequencies of neutrophils, monocytes and conventional dendritic cells (cDCs) (Appendix A). As CLL development in TCL1 mice is associated with an accumulation of patrolling monocytes [15], we quantified frequencies of monocyte subsets in the spleen by analyzing their expression of Ly6C by flow cytometry. 1MT treatment slightly reduced the frequency of Ly6C^−^ patrolling monocytes and increased the frequency of Ly6C^+^ inflammatory monocytes, therefore leading to a partial normalization of the monocyte compartment (Appendix A). Quantification of CD8^+^ effector T cells revealed a significant increase in the frequency of effector cells in 1MT-treated vs. control mice (Appendix A), and a significantly higher frequency of PD-1 positive cells within the effector population (Figure 2E and Appendix A). There was no difference in the expression of other activating or inhibitory receptors tested, namely CXCR3, CD69, CD137, CD244, lymphocyte activation gene-3 (LAG3) and CD160, or in the secretion of the cytokines tumor necrosis factor alpha (TNFα), interleukin-2 (IL-2) and interferon gamma (IFNγ) measured by flow cytometry. Even though there was no difference in the frequency of Tregs (Appendix A), 1MT-treated mice showed an accumulation of Tregs expressing CD69 (Figure 2G and Appendix A), which marks activated Tregs. Other known activation markers such as KLRG1 and GITR were not altered in these cells.

In sum, modulation of IDO1 activity with 1MT resulted in only transient effects on leukemia development in TCL1 AT mice. At the experimental endpoint, a slight reduction in tumor-supportive monocytes, as well as an increase in cytotoxic effector T cells and activated Tregs, was observed.

As epacadostat has been shown to be a more specific inhibitor of IDO1 and was tested in clinical trials for several cancer entities, we further analyzed its effects in the TCL1 mouse model. Treatment with epacadostat was started when TCL1 AT mice had an established CLL-like disease. Mice were allocated to treatment and control groups based on percentages of malignant B cells in the blood (32.3 ± 4.6 vs. 32.6 ± 5.0% in epacadostat vs. control; Figure 3A, week 0). Similar to 1MT treatment, epacadostat slightly reduced CLL expansion in blood in the majority of mice 1 week after treatment initiation, although the difference in tumor load was not significant (Figure 3A). This effect, however, disappeared again at later stages of the disease. At the endpoint, after 4 weeks of treatment, there was no difference in the spleen weight or in tumor burden in the spleen (Figure 3B,C), inguinal lymph nodes or bone marrow (Appendix A). Epacadostat treatment resulted in a similar frequency of KI67-positive, proliferating CLL cells in the spleen at the experimental endpoint (Figure 3D), which explains the minor treatment efficacy.

In the tumor microenvironment of epacadostat-treated mice, a trend towards fewer monocytes and more cDCs was observed (Appendix A). Additionally, as already seen with 1MT treatment, the frequency of Ly6C^−^ patrolling monocytes was reduced in epacadostat-treated mice (Appendix A). In contrast to mice treated with 1MT, epacadostat did not alter the frequency of CD8^+^ effector T cells (Appendix A). Whereas PD-1 expression on splenic effector T cells was only slightly decreased (Figure 3E), epacadostat treatment resulted in a significantly higher percentage of KLRG1-expressing CD8^+^ effector T cells compared to control mice (Figure 3F). As in the 1MT study, there was no difference in the expression of other activating or inhibitory receptors tested, such as CXCR3, CD69, CD137, CD244, LAG3 and CD160, or in the secretion of the cytokines TNFα, IL2 or IFNγ. In addition, the frequency of Tregs (Appendix A) and the expression of markers indicative for Treg suppressive activity, namely CD69, KLRG1 and GITR (Figure 3G,H and Appendix A), were slightly decreased in epacadostat-treated mice.

Overall, the results of these in vivo experiments point out that even though IDO1 might be involved in suppressing antitumoral CD4^+^ and CD8^+^ T cell responses, targeted blocking of IDO1, especially at the later stages of the disease, does not reverse T cell suppression or control leukemia development in the TCL1 AT model.

## 4. Discussion

An immunosuppressive involvement of IDO1 in CLL has been suggested by previous publications [8,10,20]. As we observed enhanced expression of IDO1 in tumor-associated myeloid cells in the Eµ-TCL1 mouse model of CLL, we aimed at testing the potential of therapeutically targeting IDO1 in this model. Even though we observed initial treatment effects after two weeks of treatment with 1MT, epacadostat treatment resulted in an even less pronounced efficacy at an early disease stage, and with both drugs, no major differences in tumor load in the spleen, lymph nodes, bone marrow or blood of mice were detectable at later time points.

As a tryptophan mimetic, 1MT was among the first inhibitors of Trp catabolism tested in several preclinical cancer mouse models in which moderate effects as single drug, but good efficacies when combined with a variety of chemotherapeutic agents, were observed (reviewed in [21]). These early studies also provided evidence that the activity of 1MT was dependent on T cells [22]. Later, the specificity of 1MT for IDO1 was controversially discussed as it failed to block IDO1 activity in IFNγ-treated HeLa cells as well as in protein isolates of primary human colon cancer [23]. To understand the underlying mechanism of the observed efficacy in the mouse models, several studies addressed the mode of action of 1MT. One suggested mechanism is that 1MT acts as a high-potency Trp mimetic and reverses mammalian target of rapamycin complex 1 (mTORC1) inhibition created by Trp deprivation, thereby allowing re-activation of effector T cells under immune checkpoint therapy [24]. The unexpected effects of 1MT treatment that we observed in the Eµ-TCL1 mouse model, e.g., higher PD-1 expression on CD8^+^ effector T cells and an increase in CD69 expression on Tregs, might therefore be explained by IDO1-independent effects of 1MT.

Even though the specificity of 1MT (also known as indoximod) remains elusive, its preclinical efficacy in combination with chemotherapy led to the initiation of Phase I clinical trials, in which it was well tolerated, with no increase in expected toxicities or pharmacokinetic interactions [25,26]. These studies showed that indoximod was active in a pretreated population of patients with metastatic solid tumors. Data from a single-arm Phase II trial of indoximod plus anti-PD-1 in advanced melanoma achieved an overall response rate of 56% and a complete response in 19% of patients combined with low rates of high-grade immune-related adverse events [27]. Multiple Phase II and III trials combining indoximod with other current treatment modalities, including chemotherapy, cancer vaccines and checkpoint inhibition, have subsequently been initiated. The combination of indoximod and taxane (NCT01792050), studied in a Phase II trial in patients with metastatic breast cancer, failed to meet its endpoints of progression-free survival, overall survival or objective response rate. The recruitment of patients to a Phase III clinical trial of indoximod combined with anti-PD-1 antibodies pembrolizumab or nivolumab for patients with advanced melanoma (NCT03301636) was terminated by the sponsor.

Epacadostat has been shown to be a more specific and potent inhibitor of IDO1, which forms a direct bond with the heme iron of IDO1 and thus competes with Trp for binding to IDO1 [28]. In a Phase 1 clinical trial, epacadostat has been shown to be safe with an 80–90% IDO1 inhibitory activity measured by plasma Kyn/Trp ratio [29]. Phase II/III clinical trials of epacadostat have mainly investigated its effect in combination with the anti-PD-1 antibody pembrolizumab which was generally well tolerated and had encouraging antitumor activity in multiple advanced solid tumors [30]. Phase III trials have been initiated for the treatment of many cancers including advanced melanoma, metastatic non-small-cell lung carcinoma, renal cell carcinoma, urothelial carcinoma and head and neck cancer. In the ECHO-301 trial, epacadostat + pembrolizumab was compared to placebo + pembrolizumab in patients with unresectable or metastatic melanoma. Of note, progression-free survival and overall survival did not improve by the addition of epacadostat [31], resulting in the suspension of the other clinical trials using this drug.

Trp catabolic products, such as Kyn, are known to regulate immune cells through activation of the aryl hydrocarbon receptor (AHR) (reviewed in [32]). As it has been shown that inhibition of IDO1 leads to increased AHR activity [33] which in turn induces the expression of IDO2 and/or TDO2 [34], a possible explanation for the failure of IDO1-targeted therapy in cancer is an induction of AHR activity by the inhibitor, even if Kyn formation is sufficiently blocked. In addition, we recently identified the L-amino acid oxidase interleukin-4-induced-1 (IL4I1) as a novel and very potent upstream regulator of AHR, and we showed that IL4I1 associates more frequently with AHR activity than IDO1 or TDO2 in cancer [35]. Of interest, IL4I1 is highly expressed by myeloid cells in leukemic Eµ-TCL1 mice, and we showed that these cells contribute to immune suppression in this model [15]. As we further observed that immune checkpoint blockade induces not only *IDO1* but also *IL4I1* expression, and IDO1 inhibitors do not block IL4I1 activity [35], the expression of IL4I1 may explain the failure of IDO1 inhibitors in the Eµ-TCL1 mouse model and the lack of success of clinical studies combining immune checkpoint blockade with IDO1 inhibition.

In conclusion, IDO1 as monotherapy or in combination with immune checkpoint inhibitors is not sufficient to achieve effective reactivation of antitumor activity. Our data in the TCL1 mouse model are in line with published data of clinical trials. Trp catabolism remains an attractive target for immunotherapy of cancer due to its immunosuppressive and tumor-supportive effects. Therefore, a logical next step in improving efficacies with this therapeutic approach is the development of combination treatment with inhibitors for several Trp-degrading enzymes or of drugs that target several of these enzymes in combination or AHR directly.

## 5. Conclusions

CLL is associated with an increased IDO1 expression and activity, but its pharmaceutical inhibition is not sufficient to control leukemia development in a mouse model of CLL, suggesting compensatory mechanisms, such as that of IL4I1, in place that maintain immunosuppression.

## Figures and Tables

**Figure 1 cancers-13-01899-f001:**
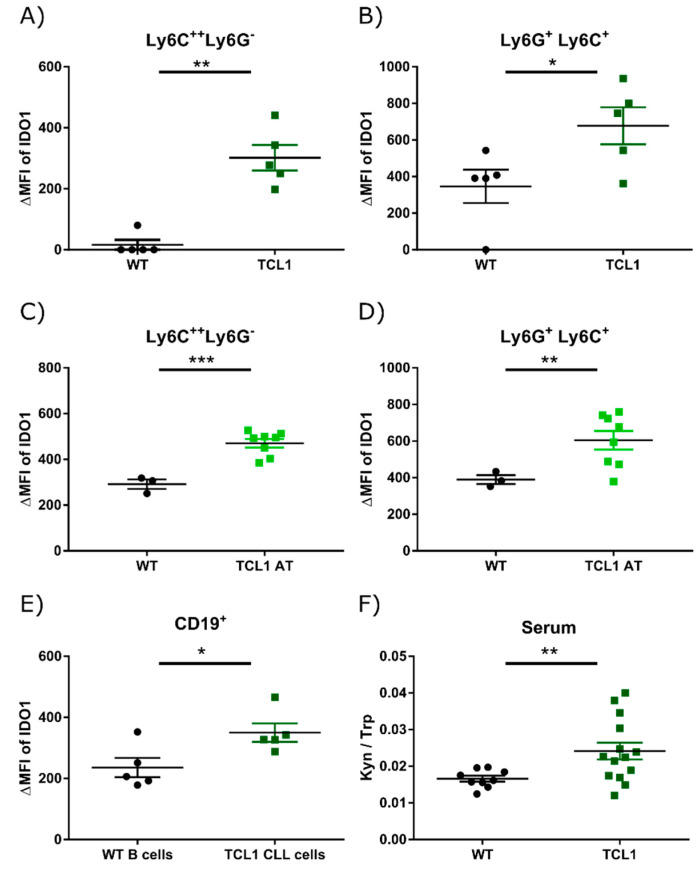
IDO1 is overexpressed in chronic lymphocytic leukemia (CLL). (**A**,**B**) Splenocytes were isolated from leukemic TCL1 and littermate wild-type (WT) mice. IDO1 expression in Ly6C-high monocytic (Ly6C^++^ Ly6G^−^) and granulocytic (Ly6C^+^Ly6G^+^) cells was determined by flow cytometry (*n* = 5 each). (**C**,**D**) Splenocytes were isolated from leukemic mice after adoptive transfer of TCL1 cells (TCL1 AT) and littermate tumor-free WT mice. IDO1 expression in Ly6C-high monocytic (Ly6C^++^ Ly6G^−^) and granulocytic (Ly6C^+^Ly6G^+^) cells was determined by flow cytometry (WT, *n* = 3; TCL1 AT, *n* = 8). (**E**) IDO1 expression in CD5^+^ CD19^+^ CLL cells from TCL1 and CD19^+^ B cells from littermate WT mice was determined by flow cytometry (*n* = 5 each). (**F**) Ratio of kynurenine and tryptophan detected in the sera of leukemic TCL1 and littermate WT mice by HPLC (WT, *n* = 9; TCL1, *n* = 1 4). All graphs show means ± SEM. Comparison of groups was performed with Welch’s *t*-test. * *p* < 0.05, ** *p* < 0.01, *** *p* < 0.001. ΔMFI = normalized median fluorescence intensity.

**Figure 2 cancers-13-01899-f002:**
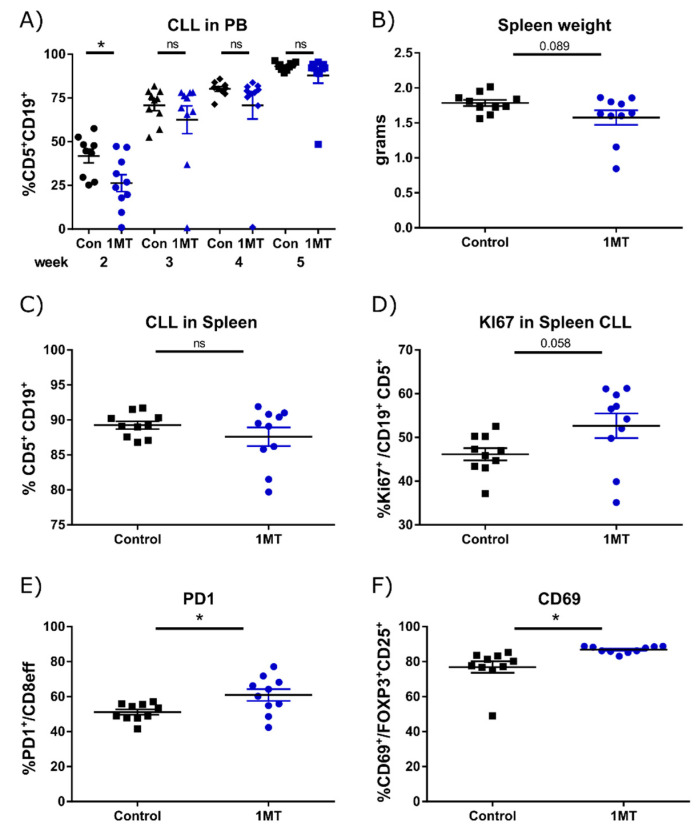
1MT treatment has minor effects on CLL development and tumor-associated immune cells. Wild-type (WT) mice were transplanted i.p. with 2 × 10^7^ TCL1 tumor cells and were divided into 2 groups (*n* = 10) directly after transplantation. Mice of the 1MT group had continuous access to the drug via drinking water. Mice of the control group (Con) received vehicle in the drinking water. (**A**) Percentage of CD5^+^ CD19^+^ CLL cells was monitored over time in peripheral blood (PB) by flow cytometry. Six weeks after tumor transplantation, (**B**) spleen weight was recorded, and (**C**) percentage of CD5^+^ CD19^+^ CLL cells in the spleen, (**D**) percentage of KI67^+^ cells within CD5^+^ CD19^+^ CLL cells, (**E**) PD-1 expression on splenic CD8^+^ effector T cells, as well as (**F**) CD69 expression on splenic CD25^+^ FOXP3^+^ regulatory T cells were quantified by flow cytometry. All graphs show means ± SEM. Comparison of groups was performed with Welch’s *t*-test. * *p* < 0.05, ns: non-significant.

**Figure 3 cancers-13-01899-f003:**
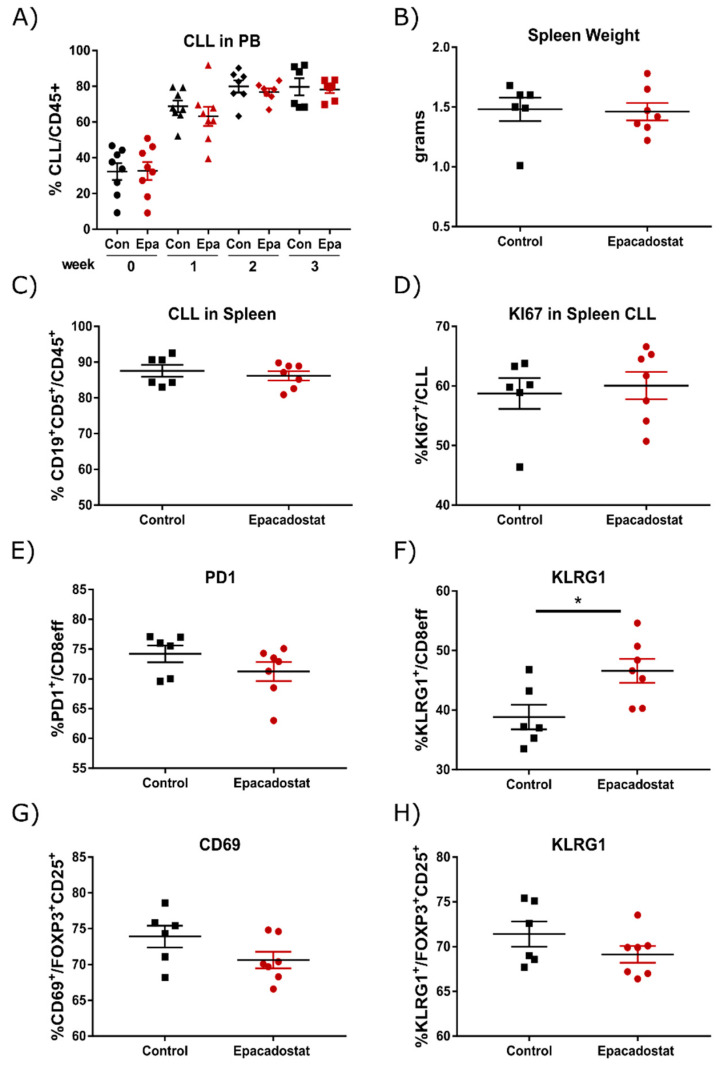
Epacadostat treatment has minor effects on CLL development and tumor-associated immune cells. Wild-type (WT) mice were transplanted i.p. with 2 × 10^7^ TCL1 tumor cells and were divided into 2 groups (*n* = 8) 2 weeks post tumor transplantation. Mice were treated with epacadostat (Epa) or vehicle control (Con) by oral gavage once daily. (**A**) Percentage of CD5^+^ CD19^+^ CLL cells was monitored over time in peripheral blood (PB) by flow cytometry. Week numbers represent the time after treatment start. Five weeks after tumor transplantation, (**B**) spleen weight was recorded and (**C**) percentage of CD5^+^ CD19^+^ CLL cells in the spleen, (**D**) percentage of KI67^+^ cells within CD5^+^ CD19^+^ CLL cells, (**E**,**F**) PD-1 and KLRG1 expression on splenic CD8^+^ effector T cells, as well as (**G**,**H**) CD69 and KLRG1 expression on splenic CD25^+^ FOXP3^+^ regulatory T cells were quantified by flow cytometry (control, *n* = 6; epacadostat, *n* = 7). All graphs show means ± SEM. Comparison of groups was performed with Welch’s *t*-test. * *p* < 0.05. Statistics are only indicated when significant.

## Data Availability

The data presented in this study are available in this article and Appendix A.

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
