# Peer review of "IDO1-Targeted Therapy Does Not Control Disease Development in the Eµ-TCL1 Mouse Model of Chronic Lymphocytic Leukemia"

_cancers, 2021, doi:10.3390/cancers13081899_

Round 1
Reviewer 1 Report
The authors describe how the IDO1-targeted therapy is not sufficient to block the tumor progression in a mouse model of CLL, despite IDO1 and its metabolites are overexpressed in Eμ -TCL1 mice compared to physiological conditions.
However, I have some questions about the experimental protocol used in this work.
I didn't understand two essential points about mice treatment (lines 121-128): the authors describe the concentration of the compound and how they are administered to mice, but it's not clear 1)for how many days the mice have drank the "supplemented" water, and 2) what volume of water the mice have assumed.
The continuous access to water and the consequent different supply among the mice could influence the slight observed variations.
It could be also useful to analyze more mice treatment groups, with different concentrations of inhibitors, to confirm that the slight observed variations are not concentration-dependent.
Furthermore, the conclusion is summarized into three sentences. I think that it could be better to fuse the Conclusion with the Discussion
Author Response
We would like to thank the referee for the critical review of our manuscript and the helpful comments.
We have added further details about the treatment of the mice under point 2.5.
“… continuous access from the first day of tumor transplantation until the end-point. Control mice received vehicle (0.1 M NaOH and 0.25 mg/ml Aspartame in drinking water). The dose and the route of administration were chosen according to previous studies showing efficacy of 1MT. …The dose and route of administration were recommended by Incyte Corporation and were similar as described by Koblish et al, 2010…”
We have not measured the volume of water the mice drank during the course of the study, however, this is the standard procedure which has been used in the literature previously (see new references 16-18) and the mice were active and seemed healthy until the endpoint and therefore consumed regular amounts of water and thus the drug.
We agree that heterogeneity within the groups can well be due to different amounts of drug consumed by the animals. However, considering the slight variations we observed and also the loss of effects over time, based on our experience with this tumor model, we believe analyzing more mice will not change the conclusion that there is no significant effect of 1MT on CLL progression.
The three-sentence-conclusions paragraph was added due to the requirements of Cancers.
Reviewer 2 Report
This manuscript by Öztürk et al. demonstrates that the inhibition of IDO1 by 1-MT (specificity in question) or epacadostat in CLL is not sufficient to control leukemia development. They utilized an Eμ-TCL1 adoptive transfer model for the current experiments. As anticipated, Figure 1 shows an overexpression of IDO1 in this CLL model. Figures 2 and 3 show the effect of 1-MT and epacadostat treatment, respectively, on CLL development and tumor-associated cells (the generalized term “tumor microenvironment” is not correct).
Overall, manuscript is well-written. However, the current descriptive and preliminary study is entirely based on the comparative observation between treated vs control mice, lacking any mechanistic detail that would add new dimension to the scientific knowledge. Contradictory data on PD1 expression shown in Figure 2E vs 3E and on CD69 expression in Figure 2F vs 3G further raise a concern on the model system. It would be great to identify the compensatory mechanism associated with increased IDO1, that may be involved in maintaining immunosuppression.
Minor comments
What are criteria for selecting 6-week or 5-week time-points and drug dosage for measurements?
What are criteria for selecting and n = 10 or 8 mice in Figure 2 or 3? Authors must clearly justify the number of mice used for the statistical power.
Include the number of technical and biological replicates for each of the data in corresponding figure legends.
Statistical significance indication is missing in some of the panels in Figure 3 and Supplementary Figures 2 & 3.
Author Response
We would like to thank the referee for the critical review of our manuscript and the helpful comments.
As recommended, we changed the term “tumor micronvironment” to “tumor-associated immune cells” throughout the manuscript.
We have recently identified a compensatory mechanism associated with increased IDO1 which involves the L-amino acid oxidase interleukin-4-induced-1 (IL4I1) that maintains immunosuppression (Sadik et al, 2020, Cell), and discuss these findings now in more detail in the manuscript.
The tendencies and the significant difference for KLRG1 on CD8 T cells upon epacadostat treatment fit to the expected improvement of immune control upon IDO1 inhibition; e.g. CD8 T cells are less exhausted (lower PD1), more activated (higher KLRG1), and Tregs are less activated (lower CD69 and lower KLRG1). However, as these changes are not strong, they are not enough to allow for tumor control. The unexpected effects of 1MT treatment (higher PD1 on CD8+ effector T cells and CD69 on Tregs) might be explained by IDO1-independent effects of 1MT, as already suggested by other studies. We have added this statement now to the Discussion part of the paper.
Minor comments:
Treatment studies were terminated when the spleen size, determined by palpation, reached approximately 1.5 cm on average in any of the groups. We have added this information to the Methods part under point 2.5.
For the treatment studies, a group size of 10 was chosen based on a simulation and on previous experience with this mouse model. For the epacadostat study, 20 mice were transplanted with tumor cells, but 4 mice had to be excluded from the experiment due to a failure of tumor cell engraftment. Therefore, treatment was performed with 8 mice per group. As 2 mice from the vehicle and 1 mouse from the epacadostat group died during the course of treatment, they could not be included for the endpoint analysis. We have added this information to the Methods part under point 2.5.
Missing numbers of replicates (all biological replicates) were added to the figure legends.
We added an explanation to the legend of Figure 3 and Suppl. Figures 2 and 3, stating that where no statistics were indicated, the difference was statistically insignificant.
Round 2
Reviewer 1 Report
Dear Authors
according to the added modifies, I think that the paper is suitable for publication in the present form
Reviewer 2 Report
I have no further comment to add.